

# Detecting telomere elongation in longitudinal datasets: analysis of a proposal by Simons, Stulp and Nakagawa

Daniel Nettle[*] and Melissa Bateson[*]

Centre for Behaviour and Evolution & Institute of Neuroscience, Newcastle University, United Kingdom
[*] These authors contributed equally to this work.

## ABSTRACT

Telomere shortening has emerged as an important biomarker of aging. Longitudinal studies consistently find that, although telomere length shortens over time on average, there is a subset of individuals for whom telomere length is observed to increase. This apparent lengthening could either be a genuine biological phenomenon, or simply due to measurement and sampling error. *Simons, Stulp & Nakagawa (2014)* recently proposed a statistical test for detecting when the amount of apparent lengthening in a dataset exceeds that which should be expected due to error, and thus indicating that genuine elongation may be operative in some individuals. However, the test is based on a restrictive assumption, namely that each individual's true rate of telomere change is constant over time. It is not currently known whether this assumption is true. Here we show, using simulated datasets, that with perfect measurement and large sample size, the test has high power to detect true lengthening as long as the true rate of change is either constant, or moderately stable, over time. If the true rate of change varies randomly from year to year, the test systematically returns type-II errors (false negatives; that is, failures to detect lengthening even when a substantial fraction of the population truly lengthens each year). We also consider the impact of measurement error. Using estimates of the magnitude of annual attrition and of measurement error derived from the human telomere literature, we show that power of the test is likely to be low in several empirically-realistic scenarios, even in large samples. Thus, whilst a significant result of the proposed test is likely to indicate that true lengthening is present in a data set, type-II errors are a likely outcome, either if measurement error is substantial, and/or the true rate of telomere change varies substantially over time within individuals.

Corresponding author
Daniel Nettle, daniel.nettle@ncl.ac.uk

## INTRODUCTION

Telomere shortening in tissues such as blood has emerged as an important biomarker of ageing (*Müezzinler, Karina & Brenner, 2013*), predictor of future morbidity and mortality (*Heidinger et al., 2012*; *Boonekamp et al., 2013*; *Rode, Nordestgaard & Bojesen, 2015*), and indicator of accumulated adversity (*Hau et al., 2015*; *Bateson, 2016*). Telomeres are repetitive DNA sequences at the end of eukaryotic chromosomes that, on average at
the population level, shorten with age. In longitudinal studies, though, there is often a substantial fraction of the sample that shows an increase in measured telomere length (*Steenstrup et al., 2013b*; *Simons, Stulp & Nakagawa, 2014*). The observation of apparent lengthening is potentially important, since it points to the possibility that a marker of cellular ageing might under some circumstances be reversible *in vivo*. However, telomere length cannot be measured with perfect precision. There is error variation both due to sampling (heterogeneity in cells within an individual lead to variable estimates of that individual's average telomere length), and measurement (laboratory assays do not produce identical results each time even with the same sample). The existence of error variation means that the second of two longitudinal samples may show a higher value than the first even if the true average telomere length has not increased. Thus, it is possible that apparent telomere lengthening in a sample represents no more than error (*Steenstrup et al., 2013b*; *Bateson & Nettle, 2017*).

*Simons, Stulp & Nakagawa (2014)* recently proposed a statistical test for detecting when there is more observed lengthening in a longitudinal sample than should be expected under the hypothesis of error alone, and hence for inferring when true lengthening is likely to be present in some subset of the sample. This is potentially a useful innovation as it might allow resolution of whether apparent telomere lengthening over time in vivo is a biologically real phenomenon or not. The test requires that each individual is measured at three or more time points. To complete the test, a ratio of two variance estimators (henceforth, the F-ratio) is compared to an F-distribution, in a similar manner to the F-test familiar from ANOVA. Under the null hypothesis (no true lengthening), the two estimators will be similar, the F-ratio will be close to 1, and the *p*-value from comparing the statistic to the F-distribution with appropriate degrees of freedom will be large (i.e., not significant). Under the alternative hypothesis (true lengthening is present), the numerator will be substantially larger than the denominator, the F-ratio will be larger than 1, and the *p*-value will therefore be small (considered significant by the usual convention when $p < 0.05$).

The numerator of the F-ratio estimates the variability in the sample by a calculation based on the number of individuals who have a higher measured telomere length at the final time point compared to the first, and the magnitude of their apparent increase (*Simons, Stulp & Nakagawa (2014)*, Eq. (5); see *Simons, Stulp & Nakagawa (2014)*, Appendix for derivation of this estimator). The denominator of the F-ratio estimates what under the null hypothesis is the same variability, in a different way. It fits a separate regression line through the points corresponding to the repeat measurements of each individual (so the number of regression lines is equal to the number of individuals in the sample). For each of these lines, it calculates the variance of the residuals, the deviations of the points from the fitted line. This is why three measurement points are required: with just two points, the line goes through both and there is no residual. Finally, the variability of the whole sample is estimated as the mean of the residual variance from each of the separate individual regressions (see *Simons, Stulp & Nakagawa (2014)*, Eqs. (1)–(3)).

There is an important assumption involved in the specification of the denominator of the F-ratio statistic, namely that each individual's telomeres truly change at a constant rate over time. Thus, any deviation of the individual's successive measurement points from
a straight line (either going up, going down, or flat) can be taken to represent sampling or measurement error. However, it is not currently known whether this assumption is empirically plausible or not. The pace of telomere shortening has been linked to infection (*Asghar et al., 2015*), adverse life events and stress (*Epel et al., 2004*; *Puterman et al., 2014*), and health behaviours (*Puterman et al., 2014*). All of these factors are episodic or changeable over time, so it could be that individuals' telomeres change at different rates—or even in different directions—in different years, without this being in any sense due to measurement or sampling error. Two recent papers have specifically suggested that telomeres shorten in a dynamic or oscillatory way, in which one year's true attrition is not predictable from the previous year's (*Svenson et al., 2011*; *Huzen et al., 2014*).

If there are year-to-year changes in individuals' rate of true shortening, then the linear regressions for each individual would not fit perfectly, even if telomere length could be measured with no error at all. The denominator of the F-ratio statistic proposed by *Simons, Stulp & Nakagawa (2014)* thus actually sums together two components: the variability over time of the *true* rate of telomere change within individuals, plus the measurement and sampling error. This means that, where there is any variability in individual shortening rates over time, the denominator of the test will be larger than it should be for the purposes required of it, the F-ratio will consequently be too small, and the test will potentially produce a high rate of type-II errors (that is, false negatives, or failures to return a significant result when a substantial proportion of the population do exhibit true lengthening each year).

It is common for statistical tests to rely in their derivation on assumptions that are not exactly met in real phenomena, but yet the tests still turn out to be useful. Thus, the question is, how large would departures from constant rates of true shortening have to be to cause substantial problems of type-II error for the proposed test? This question interacts with the extent of measurement error. *Simons, Stulp & Nakagawa (2014)* show in simulations that, other things being equal, increasing the extent of measurement error reduces the power of the proposed test. This may be particularly true if the constant-true-rate assumption is also violated. Here, we simulated large longitudinal telomere datasets, systematically varying the extent of measurement error (none, small, large), and the assumed underlying true dynamics (constant true rate for individuals, no individual consistency in the true rate, moderate individual consistency in the true rate). To maximise the relevance to empirical applications of the test, we used reported values from the human telomere literature in constructing our simulations. Our objective was to estimate the likely power of the test to detect true lengthening when true lengthening is in fact present. We kept the sample size in our simulated datasets at 10,000 individuals throughout, so as to be able to understand the power of the test even as sample size becomes very large.

## METHODS

Our simulations are based on a computational model described formally in the Appendix, and explored more fully in *Bateson & Nettle (2017)*. The R code to generate all the results that follow is available as Supplemental Information 1. The model assumes that telomere length is measured every year, and it can be iterated to give as many years of data as required.

In the first stage of the model, the true telomere lengths at each time point for $n = 10,000$ individuals are generated. The baseline telomere lengths are drawn from a normal distribution with mean 7,000 base pairs (bp) and standard deviation 700 bp. The second year's telomere lengths are generated by subtracting a normally distributed random amount with mean 30 bp and standard deviation 50 bp. This means that although the average telomere length shortens from baseline to the second year, some individuals truly lengthen. For example, an individual whose attrition is one standard deviation from the mean in the positive direction actually experiences lengthening of 20 bp. The values for the means and standard deviations of baseline telomere length and attrition are representative of the empirical human literature (*Aviv et al., 2009*; *Chen et al., 2011*; *Kark et al., 2012*; *Steenstrup et al., 2013a*).

In each subsequent year, attrition is repeated, again with a mean of 30 bp and standard deviation of 50 bp. Attrition in each successive year can be made to be correlated with attrition in the previous year (each new year's attrition values are generated from the last using Eq. (5) of Appendix). The level of autocorrelation is controlled by a parameter $r$. In the case where $r = 1$, the amount of telomere change, whether shortening or elongation, is constant from year to year. Thus, the $r = 1$ case captures the assumption made by *Simons, Stulp & Nakagawa (2014)* in the derivation of their statistic. Where $r = 0$, attrition is completely independent from year to year; an individual with relative fast attrition in one year is just as likely as any other to have slow attrition the next year. Here, we investigate three values of $r$: $r = 1$, where *Simons, Stulp & Nakagawa*'s *(2014)* constant-rate assumption holds; $r = 0$, where there is no individual consistency at all in the rate of telomere change; and $r = 0.5$, where there is partial but not complete individual consistency in the rate of change over time, and so *Simons, Stulp & Nakagawa*'s *(2014)* assumption may be useful as an approximation.

In a second stage of the model, measurement error can be introduced by assuming that measured telomere length at each time point is an independently generated random sample from a normal distribution with the mean equal to the true telomere length. For the standard deviation of this error distribution, we investigated three values: 0, i.e., no measurement error; 140 bp; and 560 bp. The latter two values were chosen to be high and low in the range of recent estimates of the magnitude of technical variation in telomere measurement (98–665 bp; *Martin-Ruiz et al., 2015*; *Bateson & Nettle, 2017*). Note that measurement error is implemented as a fixed standard deviation around the true length, and not as a coefficient of variation as in our previous paper (*Bateson & Nettle, 2017*). Recent evidence suggests that the assumption implicit in the construction of a coefficient of variation (that measurement error is proportional to the telomere length measured) may not hold for telomere measurement, at least when done by qPCR (*Verhulst et al., 2015*).

We used the model to generate one hundred datasets at each combination of: two to eleven years of follow-up; autocorrelations of $r = 1$, $r = 0.5$ and $r = 0$; and the three levels of measurement error. All of these datasets contained true telomere lengthening, though the proportion of true lengtheners varied as functions of both length of follow-up and autocorrelation (*Bateson & Nettle, 2017*). For each dataset, we calculated the F-ratio statistic using the code provided by *Simons, Stulp & Nakagawa (2014)*. We investigated, for

each combination of years of follow-up and $r$: first, how many true lengtheners there were in each dataset; and second, how many of the possible 100 F-ratio tests were significant by the conventional criterion of $p < 0.05$.

## RESULTS

In Fig. 1, the points and dashed lines show the proportion of times the F-ratio test proposed by *Simons, Stulp & Nakagawa (2014)* produced a significant result, as a function of the number of years of follow-up, and broken down by the autocorrelation of individuals' annual true telomere attritions ($r = 0$, $r = 0.5$ or $r = 1$), and the level of assumed measurement error (SDe = 0, SDe = 140, SDe = 560). The mean proportion of individuals whose telomeres truly lengthen varies as a function of $r$ and the length of follow-up; it is shown as the solid line in each panel of Fig. 1. The grey area shading corresponds to regions where more than 5% of individuals showed true telomere lengthening, and so it would be desirable for the proposed test to return a significant result.

We first consider the case where there was no measurement error (Figs. 1A, 1D and 1G). Where *Simons, Stulp & Nakagawa*'s *(2014)* assumption of a constant true rate was met (Fig. 1G), the test successfully returned a significant result for every dataset using these large samples. The same was also true when the constant-true-rate assumption was not exactly met, but there was moderate temporal consistency in the true rate (Fig. 1D). However, when there was no individual consistency in the true rate of attrition (Fig. 1A), the proposed test systematically returned type-II errors for follow-up periods of five years or more, even with no measurement error.

The second column of Fig. 1 shows the case of measurement error equal to a standard deviation of 140 bp. Here, the test had low power (under 0.25) when *Simons, Stulp & Nakagawa*'s *(2014)* assumption of a constant true rate was met (Fig. 1H), even in these samples of 10,000 individuals. Where the assumption was not met (Figs. 1B and 1E), the test always returned a non-significant result. Finally, we considered measurement error equal to a standard deviation of 560 bp (Figs. 1C, 1F and 1I). Here, the test always returned a non-significant result, although substantial fractions of the population exhibited true lengthening.

## DISCUSSION

We considered the performance of the F-ratio test proposed by *Simons, Stulp & Nakagawa (2014)* on simulated longitudinal datasets, under different scenarios for the nature of the true telomere dynamics and the magnitude of measurement error, where there was a non-zero and known proportion of true telomere lengtheners, and the sample size was very large. Ideally the test should have been significant in all or the vast majority of cases, particularly those where the proportion of true lengtheners was substantial. We found that, whilst the test correctly detected lengthening under two of our nine scenarios, for the remainder, it either always or usually returned a type-II error. That is, it led to the acceptance of a null hypothesis (no true lengthening) that should have been be rejected.

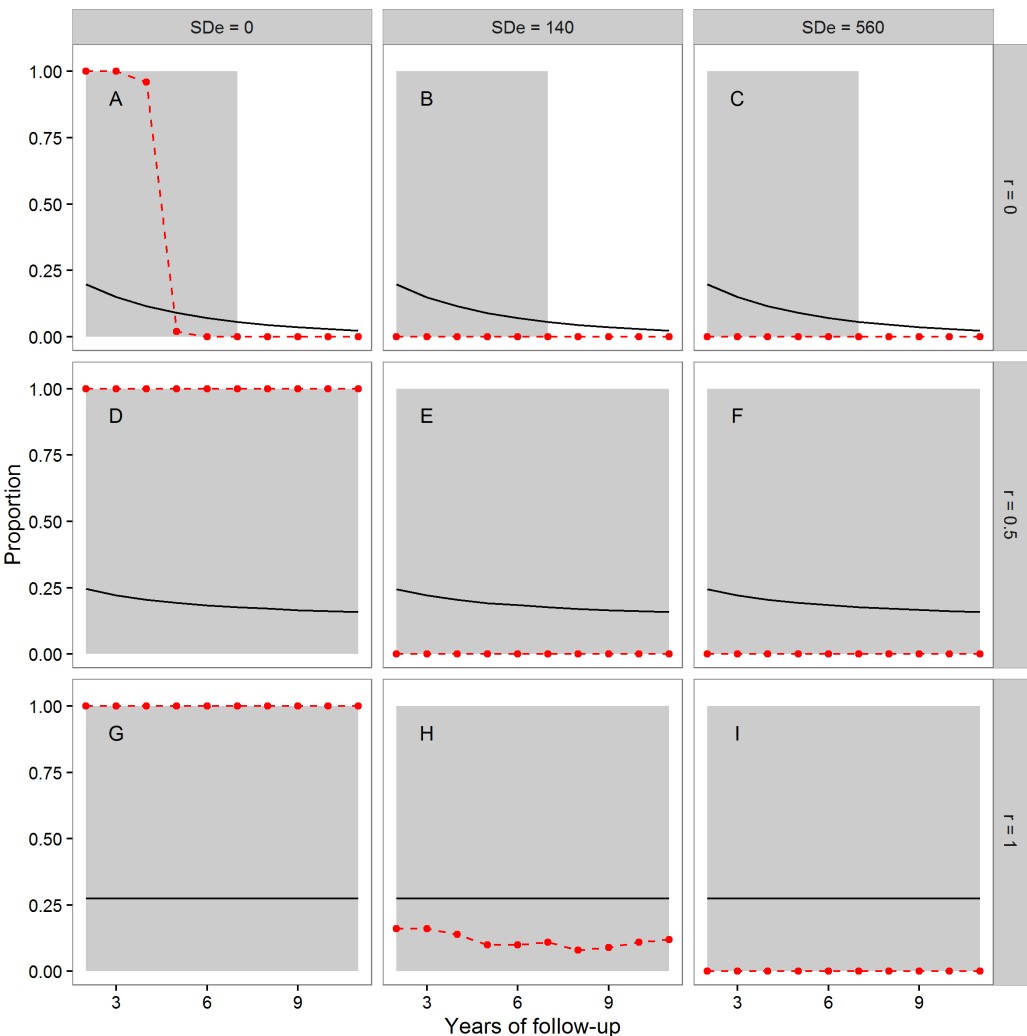

**Figure 1** The mean proportion of individuals exhibiting true telomere lengthening (solid line), and the proportion of times the F-ratio test proposed by *Simons, Stulp & Nakagawa (2014)* returned a significant result (points and dashed lines), for different numbers of years of follow-up, split by values of the autocorrelation parameter $r$ (A, B, C: $r = 0$; D, E, F: $r = 0.5$; G, H, I: $r = 1$), and level of measurement error (A, D, G: 0 bp; B, E, H: 140 bp; C, F, I: 560 bp). The first point is after two years of follow-up, since this is the earliest point where the test statistic can be calculated (baseline plus two follow-up measurements). The grey area shading covers regions where the proportion of the population exhibiting true lengthening is greater than 5%. When $r = 1$, individuals have a constant rate of change over the whole time period. When $r = 0$, an individual's telomere change in one time period is independent of their change in the previous period. $r = 0.5$ indicates moderate individual consistency in the rate of change. At each combination of $r$, measurement error, and years of follow-up, 100 datasets each of 10,000 individuals were simulated.

Our first conclusion is that measurement error at the levels that have been reported in the human literature reduces the power of the proposed test to a low level. Under our smaller and larger non-zero measurement-error scenarios, the test returned a non-significant result almost all of the time. This was despite our using samples (10,000 individuals) that are at the upper end of the size range studied in practice by empiricists.

The finding that increased measurement error reduces the test's power accords with the power simulations presented by *Simons, Stulp & Nakagawa (2014)*. They found that power was good as long as the standard deviation of true attrition was larger than the standard deviation due to measurement error (see *Simons, Stulp & Nakagawa (2014)*, Fig. 1). We agree, but would argue that the standard deviation of attrition is generally much smaller than the standard deviation due to measurement error in practice. For humans, the best empirical estimates are that the standard deviation of annual true telomere attrition is of the order 14–53 bp/year for humans (*Aviv et al., 2009*; *Chen et al., 2011*; *Kark et al., 2012*; *Steenstrup et al., 2013a*), whilst the standard deviation due to measurement error is of the order of 98–665 bp (*Martin-Ruiz et al., 2015*; *Bateson & Nettle, 2017*). Technical precision may vary from technique to technique (*Verhulst et al., 2015*; *Verhulst et al., 2016*), and running extra technical replicates can reduce the magnitude of measurement error (*Verhulst et al., 2015*; *Eisenberg, 2016*). Nonetheless, researchers using the test should be mindful that if the magnitude of the measurement error in their data is the same as or larger than the magnitude of the variation in true telomere attrition, the test will be prone to return type-II errors.

Our second conclusion concerns *Simons, Stulp & Nakagawa*'s *(2014)* assumption that the true rate of telomere attrition is perfectly consistent within individuals over time. Violations of this assumption also reduce the power of the test. In particular, the test never once returned a significant result, in 4000 attempts, where the constant-true-rate assumption was not true and there was any measurement error. Even with no measurement error, the power of the test was very low at long follow-ups under the scenario of no individual consistency in true attrition from year to year. These type-II errors are understandable. When the true rate of attrition varies within individuals, the denominator of the F-ratio is systematically too large, because it adds the variability in the annual rate of true attrition to the calculation of the error variation. Thus, the F-ratio statistic is almost always less than one, and a significant result can very rarely be generated. Thus, the *Simons, Stulp & Nakagawa (2014)* approach to detecting telomere elongation would be problematic if it turned out that the true rate of attrition varies substantially from year to year.

We do not currently know to what extent individuals' true telomere losses are consistent from year to year. *Bateson & Nettle (2017)* used observed patterns of apparent lengthening in data sets with different durations of follow-up to estimate that individual consistency in the rate of attrition is low. Two recent empirical studies have suggested that telomere change tends to oscillate, with periods of rapid attrition followed by periods of elongation (*Svenson et al., 2011*; *Huzen et al., 2014*). The issue is far from settled, though, and there have not been systematic attempts to distinguish fluctuation in true dynamics from measurement error in longitudinal data. However, given the uncertainty about the extent of individual consistency, it does seem somewhat restrictive to base a test on the assumption that the individual consistency is perfect. Indeed, what attracts researchers to telomere length as a biomarker is precisely that the rate of attrition seems to vary in relation to life events (*Epel et al., 2004*; *Shalev, 2012*; *Asghar et al., 2015*; *Bateson, 2016*). Thus, the interpretation of a non-significant result from the *Simons, Stulp & Nakagawa (2014)* test, even in a very large sample, should be cautious.

Although we argue that the proposed test is likely to suffer from low power, we do not have a simple correction or an alternative test to propose. This is because basic questions about the nature of telomere dynamics over time remain unanswered, and answers to these questions are required in order to ground any test in appropriate assumptions. The most relevant question in the current context is whether there is individual consistency in the rate of telomere shortening; and if, so, whether this arises from consistent environmental influences, developmental factors, or genetic variation. Telomere dynamics are likely to vary between species, and so different models of how telomeres change may be appropriate to different systems. Our simulations with moderate but imperfect individual consistency generated the consistency through an autoregressive process of order one; this is not the only possible method, and may not be the most appropriate. Thus, we would appeal to the field to conduct large longitudinal studies with more than two measurement time points. As well as shedding light on the appropriateness of *Simons, Stulp & Nakagawa*'s *(2014)* true-constant-rate assumption, this would help us to build better process models of how telomeres change, and hence to derive robust statistical models against which empirical data can be compared.

## APPENDIX

For each individual in each dataset, baseline telomere length in base pairs is generated by:

$$length_b \sim N(7000, 700) \tag{1}$$

Length at the first follow-up year is then generated by:

$$length_1 = length_b - attrition_1 \tag{2}$$

$$attrition_1 \sim N(30, 50) \tag{3}$$

For all subsequent years:

$$length_{y+1} = length_y - attrition_{y+1} \tag{4}$$

$$attrition_{y+1} = r \cdot attrition_y + \sqrt{(1-r^2)} N\left(\frac{(1-r)}{\sqrt{(1-r^2)}}30, 50\right) \tag{5}$$

Equation (5) generates attrition values that have the required level of autocorrelation $r$, whilst maintaining a mean attrition of 30 bp and a standard deviation of attrition of 50 bp (for proof see *Bateson & Nettle, 2017*).

Finally, measurement error is added to all telomere lengths using:

$$measured_y \sim N(length_y, SD_e) \tag{6}$$

Here, $SD_e$ represents the magnitude of measurement error, taken as either 0 bp, 140 bp or 560 bp, as specified.

### Funding

This research was supported by the National Centre for the Replacement Refinement and Reduction of Animals in Research (NC3Rs) under grant number NC/K000801/1 and the European Research Council (ERC) under grant number AdG 666669 (COMSTAR). The funders had no role in study design, data collection and analysis, decision to publish, or preparation of the manuscript.

### Grant Disclosures

The following grant information was disclosed by the authors:
National Centre for the Replacement Refinement and Reduction of Animals in Research (NC3Rs): NC/K000801/1.
European Research Council (ERC): AdG 666669.

### Competing Interests

The authors declare there are no competing interests.

### Author Contributions

- Daniel Nettle conceived and designed the experiments, performed the experiments, analyzed the data, wrote the paper, prepared figures and/or tables.
- Melissa Bateson conceived and designed the experiments, performed the experiments, analyzed the data, reviewed drafts of the paper.

### Data Availability

  The raw data has been supplied as a Supplementary File.

### Supplemental Information

Supplemental information for this article can be found online at http://dx.doi.org/10.7717/peerj.3265#supplemental-information.

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
