# Peer review of "Detecting telomere elongation in longitudinal datasets: analysis of a proposal by Simons, Stulp and Nakagawa"

_PeerJ, doi:10.7717/peerj.3265_

## Round 0.1 · original submission · Major Revisions

Science progresses by critical evaluation. This paper does this. The question is whether the criticism is valid. A minimum requirement for a statistical significance test is that it maintains or controls the type I error rate (the false positive rate) under the null hypothesis. Control is even weaker than maintain: it does not give higher rates than the nominal rate, e.g. 5%, but may give lower rates). The null hypothesis in SSN is on the population level: no individual shows TL increase. This situation is not simulated in the current study (see line 148) and thus not evaluated. It cannot be concluded therefore that the SSN test fails (line 36). What remains is that the test (under the set parameter values) is shown to have low power (e.g. 25% with 3 year follow-up and 25% true lengtheners).

The paper does not attempt to propose a more powerful test (that still controls the error rate) whereas this should not be too difficult. In this sense the paper is only just above the minimum publishable unit in my view.

It would also be good to comment on the power analysis in SSN, e.g. is it statistically sound and in which way did it differ from yours?

From the points raised by Mirre Simons, you should address particularly the questions on the parameter settings of the simulation. Simulations with constant error (instead of CV) would welcome (e.g. report those in Supplement).

Additional details
L101 What does fail mean here? See General. See https://peerj.com/articles/2885/
for a test in another context that failed.
L128 Add references.
L152 Where does the 100 come from?
L161 Lower line -> solid line?

I downloaded the R file in the Suppl Mat, but it was not a text/R file file.

Additional comments on SSN

1. The numerator of the F test is usually derived and calculated under the alternative hypothesis. This is actually done in the paper (equation A10) but in the main text and further on in the Appendix it is said otherwise. This (illogical) aspect does have consequences for the tests.
2. The Appendix describes a mixed model (two-level regression) but this model is not exploited in the test.
3. The selection of individuals with measured increase between first and last measurement appears strange to me/a statistician.
4. The numerator does not use all the data, except perhaps in the case with three time points per individual which may deserve special attention.

·

Basic reporting

The paper is written in a very clear and concise manner. The authors explain the context and used mathematical formulas extremely well. The background is stated effectively and sufficiently.
The article is well structured and the figure is clear and informative. The authors shared their R script which is well documented.
The purpose of this paper was clearly stated: The authors sought to test a statistical test that was developed by Simons, Stulp and Nakagawa (2014) (SSN) to investigate a dataset for the presence of individuals that truly lengthen their telomeres in contrast to apparent telomere lengthening that is completely due to measurement error. The results of the present paper show clearly that the assumptions of SSN's test are too restrictive and unlikely to be met by a biological data set.

Experimental design

The authors presented original primary research within the scope of the journal.
To test further if observed telomere lengthening in a number of data sets is genuine or just due to measurement error is an important question to ask, because it might change our understanding of telomere length dynamics over life. The test proposed by SSN would be a useful tool, but it assumes a constant change of telomere length within an individual and Daniel Nettle and Melissa Bateson challenge that assumption, because it is not supported by the scientific literature. Actually the contrary is true with different changing rates found in connection with different life styles or the onset and progression of diseases. The authors of the present study, therefore, use simulated data to test the statistic by SSN under more realistic circumstances: Rate and direction of RTL change are allowed to vary.
The methods have been clearly described and the underlying model has been published before. The authors provided their R script and therefore not only make it possible to replicate their results, but also to adapt the script for its application to different scenarios.

Validity of the findings

Data is robust, statistically sound and controlled.

Conclusion is well stated.

Additional comments

I really enjoyed reading your paper and barely found anything to criticise. I have a few minor points I would like to raise, because they might improve your manuscript even further:
1) In line 48 you define telomeres as structures that shorten with age. It sounds like a general undeniable feature of the trait. Yet, your paper is about accepting the idea that true lengthening can happen. Even though you make it clearer in the following sentence, I would add “on a population level”, “on average” or “in cross-sectional studies” to the end of the sentence in line 48.
2) In line 75 you are suddenly using “SSN” as abbreviation, without defining it in the text above. You could add “(SSN)” behind the reference in line 60.
3) In the figure capture I would consider moving the explanation of the solid line (last sentence) to line 274. Then you would answer the main question readers might have (“What are the two different lines?”) in the beginning of the caption and it is easier to then focus on differences between facets.
4) In your R-script you kindly warn that a line takes a long time to run. Would it be possible to add what “long” might mean for an average computer (or your computer)? Do you think hours is long or days or weeks?

·

Basic reporting

all fine, except if included in specific comments below and in box 3.

Experimental design

Integrated in box 3.

Validity of the findings

The authors simulate and test a method we published a while back, to detect true telomere elongation from error in longitudinal datasets. I very much appreciate the effort that has gone into this work and the interest in our work and the telomere elongation problem. There are a few serious issues I feel however. One concerns biology, the others concern statistical issues with the simulations. Also, other specific comments below. In conclusion, I feel that it will be premature to strongly dismiss the only method to formally statistically test for telomere elongation very strongly as the authors do. I do appreciate the efforts here of course, but I just feel that the conclusions are very permissive especially considering the major issues I have identified below.

Biology:
In our original work, we assumed a constant true telomere attrition between individuals. As the authors now argue this might not have been realistic as telomere length is also susceptible to environmental effects. The literature is quite divided on this topic however. See for one example:
Benetos A, Kark JD, Susser E, Kimura M, Sinnreich R, Chen W, et al. Tracking and fixed ranking of leukocyte telomere length across the adult life course. Aging Cell. 2013 Jul 31;12(4):615–21. ++++++++++++ Which argues that telomere attrition in humans is very constant with correlations between follow up and baseline of r= 0.9, at the very upper range of what is now modelled in this paper.
Whether these environmental conditions transiently affect telomere attrition needs to be confirmed, as well, it might have an effect on telomere attrition rate that is not reversible. There might be a shift in TL attrition that is then insensitive to other environmental perturbations. Note that part of our error assumption also includes such biology, it just assumed this is deviation from a set value of telomere length at each timepoint and biological changes in cell populations for example that are random and normally distributed should be captured and estimated by our method.

Also, note that the biology included to parametrise the models the authors used here, but published in Aging Cell in 2016, use parameters that are subject to measurement error (such as the autocorrelation between successive measurements, which is very low for gel based methods and high for qPCR based methods..). The variance in slopes is potentially larger or smaller depending on corrections for such measurement error. It will be hard, but an interesting challenge, to dissect the temporal variation in telomere attrition and an underlying general propensity for the organism (or tissue) to show a set telomere attraction with age. The authors now make a complicated assumption on biology however that I think has little support (currently) and is debatable.

Moreover, how the authors now model the underlying telomere biology is a transient change in attrition rate that is stochastic. This seems like quite a strong biological assumption for which the authors have no evidence at this time or at least do not provide it (I have not seen it, and the authors do not cite such evidence). The model assumes there is a set attrition rate per t to t+1, which determines the next attrition rate at t+1 to t+2, via an autocorrelation (which is assumed to by a consequence of environmental effects on the biology). This will mean some individuals through stochastic processes will reduce in attrition just because their first draw on which the autocorrelation act was lower. This assumes that past telomere attrition infers the next time point. This might reflect biology, but it might not as well (as the authors also admit lines 219-226). Our, I think, more parsimonious assumption (but this is I guess up for debate) was that there is a set telomere attrition per individual with set stochastic error at each time-point.

The assumptions of true lengthening therefore remain speculative I think, and therefore to conclude this model is parameterised to actual biology might be misleading, and might therefore falsely discredit the only current solution to investigate telomere lengthening statistically.

Conclusion on biology and interpretation:
Of course, I agree, as with any test, if you do not reject your null hypothesis you cannot assume you have proved the null hypothesis. We have not claimed this is the case, we have claimed that under out biological assumptions (which the authors might disagree with, although I think this is debatable) the test can perform reasonably well and one could detect statistical evidence for true lengthening. As the authors also write: a significant test does provide evidence for true lengthening (line 232), a failure to reject the null hypothesis is hard to interpret (which is the case with all non-informed statistics, except if one would use informative priors in a Bayesian framework). The authors go on to write that the test is unlikely to have any practical application (line 238), which I think is unfair. Future will tell when we learn more about the biology of telomere attrition. The current state of the field is that sometimes lengthening is just dismissed as measurement error (without any test, an underpowered test we developed is better than none!) or is falsely interpreted to be true without any statistical scrutiny. I must say in that respect our test does have a practical use.

In the last paragraph, the authors state users should use their data to optimise the model they present in their Aging Cell 2016 paper. Key here is however that that model makes assumptions about the biology (as we do for our power analyses as well) that might be false, and hence these could skew interpretation strongly. This should be mentioned. I have suggested before and I think we might have alluded to this in our original paper that the methods we outline will be applicable to a mixed model context in which random slopes are estimated and autocorrelations can also be estimated if one would wish. This would generate a statistical testing framework of biology as well. The statistics in a mixed model context will need to be formalised and simulated to test their functionality, but this should be possible. Such an exercise will probably be more fruitful than numerical optimisation of a set model describing specific biology of telomere attrition, as the authors suggest.

Statistics:
The authors use the estimates of error of telomere measurements from Martin-Ruiz et al 2014. There are major issues with this paper, and it is clear that measurement error can be a lot lower. Such a lower error will falsify the claims made that the test we developed earlier is impractical because deviation in TL attrition will always be lower than actual error, which might not be the case (line 189). In my view, it is questionable how useful it is to parameterise telomere data as there is a lot of differences between studies in these estimates due to different techniques used to estimate TL (qPCR versus gel based methods is just one). Given the statistical problems outlined in the references below with the main paper used to parameterise their models, I feel such parameterisation might be given too much value and should not be used to dismiss any attempts to formally test for elongation (as we do with our method the authors criticise). Also, the authors based their parameterisation on human data mostly whereas the biology of telomeres is different between species and hence dismissing our method by using human data (with its inherent problems as outlined above) will be premature as our method has been used in other systems (manuscripts in prep, not by me, but others).

We have written two statistical commentaries on Martin-Ruiz et al 2014 (and others) that the authors will need to incorporate:
-Verhulst S, Susser E, Factor-Litvak PR, Simons MJP, Benetos A, Steenstrup T, et al. Commentary: The reliability of telomere length measurements. Int J Epidemiol. 2015 Oct;44(5):1683–6. 
-Verhulst S, Susser E, Factor-Litvak PR, Simons M, Benetos A, Steenstrup T, et al. Response to: Reliability and validity of telomere length measurements. Int J Epidemiol. Oxford University Press; 2016 Aug 1;45(4):1298–301. 

The authors use CV to calculate the error on telomere length. This will bias error to be larger at larger means. We have shown how this can be a problem for other reasons as well in the two commentaries cited above (and in the responses on this actually others have argued the same). For this work, specifically, however the use of CV is very problematic to the degree that could negate some of the findings the authors report on the power calculations. The use of CV will bias error to higher means of TL, as the sd calculated to draw the error from is dependent on the mean. Hence the individuals that show lengthening, with thus higher means, will have substantially higher error (the magnitude of this effect will be determined by the sd of TL in the population), and true elongation will thus be a lot harder to estimate. Also the residual method we use to get the measurement error assumes random normally distributed error (which can be checked by the user of our methods, and we make clear this is an assumption), which in this case will not be true as error will be correlated to the mean.

Comparison autocorrelation r=1 with our power analyses. It is clear the power analyses differ substantially, between our work and that presented here. My suspicion is that the erroneous use of CV will have contributed to this (see above). Also it has to be clear that this test case is with a parameterised model to human biology that might be debatable (see above) and it would be valuable to have a wider range or error versus slope variance as we did in our original paper. Actually we purposefully did not parameterise the simulated data we used for our power calculations in 2014 as it is the relative difference between error at every timepoint and the variance in slope that determines power (as the authors also observe and write in their manuscript now). Therefore, without knowledge of the relative variance of telomere lengthening such power analyses are only a proof of principle and a guide to the relative magnitude of effects that could be detected. I do not think the simulations by the authors as presented now (even with errors I outlined above corrected) could go far beyond this, without any additional inference from biology. Again, I feel that it is premature to dismiss the only (and I stress only) method available on the grounds of a power analysis that is parametrised on uncertain biology. Even if one assumes such parameterisation is quite certain, I think it will still be worthwhile to use the test, but knowing type II error is potentially large. Many formal tests are used that suffer from high type II error because alternatives are lacking.

Additional comments

-You write cellular ageing at line 52, as if telomeres are causally related to ageing and that therefore one might wish to reduce it. I disagree with that interpretation:
Simons MJP. Questioning causal involvement of telomeres in aging. Ageing Res Rev. 2015 Nov 1;24(Pt B):191–6. 
-Please use italics for in vivo
-line 152. I would do at least 1,000 tests to get a estimate of mean power from simulations.
-Line 127-128. This needs rephrasing I think (see above) and also needs to say that it will be hard to get true values as all will be subject to measurement error, especially for error prone methods such as qPCR.
-Line 171. True lengtheners yes, but the biological interpretation differs between your model and ours, see above.
-Line 194. Martin-Ruiz’s paper is problematic, see above.

---

## Round 0.2 · Minor Revisions

Thank you for addressing the points in the revision. Also, the R code in the supplement ran without error on my PC. I have two minor points.
line 37 "returns type II errors" add something in the spirit of: (false negatives, i.e. the test does not detect lengthening where a considerable fraction of the population does show lengthening).
line 103: see remark for line 37.
line 155: add the measurement error scenarios.

---

## Round 0.3 · accepted · Accept

All is OK now. The staff will make sure the abstract will be as in the paper docx, I hope.